# Internet-based peer support interventions for people living with HIV: A scoping review

Stefanella Costa-Cordella[1,2,3], Aitana Grasso-Cladera[1,3], Alejandra Rossi[3], Javiera Duarte[1,2], Flavia Guiñazu[4], Claudia P. Cortes[5,6]*

1 Centro de Estudios en Psicología Clínica y Psicoterapia (CEPPS), Facultad de Psicología, Universidad Diego Portales, Santiago, Chile, 2 Instituto Milenio Depresión y Personalidad (MIDAP), Santiago, Chile, 3 Centro de Estudios en Neurociencia Humana y Neuropsicología (CENHN), Facultad de Psicología, Universidad Diego Portales, Santiago, Chile, 4 Web Intelligence Centre, Facultad de Ingeniería Industrial, Universidad de Chile, Santiago, Chile, 5 Hospital Clínico San Borja Arriarán & Fundación Arriarán, Santiago, Chile, 6 Departamento de Medicina, Facultad de Medicina, Universidad de Chile, Santiago, Chile

* cpcortes@uchile.cl

## Abstract

Peer support interventions for people living with HIV and AIDS (PLWHA) are effective, but their associated time and material costs for the recipient and the health system make them reachable for only a small proportion of PLWHA. Internet-based interventions are an effective alternative for delivering psychosocial interventions for PLWHA as they are more accessible. Currently, no reviews are focusing on internet-based interventions with peer support components. This scoping review aims to map the existing literature on psychosocial interventions for PLWHA based on peer support and delivered through the internet. We conducted a systematic scoping review of academic literature following methodological guidelines for scoping reviews, and 28 articles met our criteria. We summarized the main characteristics of the digital peer support interventions for PLWHA and how they implemented peer support in a virtual environment. Overall the reported outcomes appeared promising, but more robust evidence is needed.

## Introduction

Human Immunodeficiency Virus (HIV) affects more than 37.7 million people worldwide and its prevalence is still increasing [1]. The primary HIV treatment is Antiretroviral Therapy (ART) which works to suppress replication of the virus resulting in improved immune response and reduced viral load. However, inadequate adherence to ART is associated with morbidity and mortality [2–4].

Because of the essential role of adherence in the success of ART, a myriad of research has been carried out to understand ART adherence. Among factors that predict HIV treatment adherence, an important role has been found in psychosocial factors such as social support [5–7], HIV stigma [6, 8, 9], stress and depression [7, 10–14], violence [15, 16] and alcohol and other drug consumption [17–19], which increase the probability of a disadvantageous outcome by adding substance abuse stigma [20]. Consequently, different psychosocial interventions have been developed to address treatment adherence, and they have increasingly been

**Data Availability Statement:** All files are available from the OSF database (osf.io/5bkzv).

**Funding:** This study was funded by the Chilean National Agency of Research and Development

(Agencia Nacional de Investigación y Desarrollo de Chile) through FONDEF to CC (ID20I10174), and the Chilean National Agency of Research and Development (Agencia Nacional de Investigación y Desarrollo de Chile) through FONDECYT to AR (N° 1190610). The funders had no role in study design, data collection and analysis, decision to publish, or preparation of the manuscript.

**Competing interests:** The authors have declared that no competing interests exist.

demonstrated to enhance HIV adherence and improve health in people living with HIV/AIDS (PLWHA) [10, 11, 19, 21–26]. Peer support is the support provided by people who share life experiences [27]. Applied to interventions, peer support typically includes group meetings, support networks (either virtual or in-person), or peer-mentoring [28]. Peer support has been a common and effective strategy for people living with stigmatized conditions [29–32]. Peer support is also efficient in lowering the overall costs of medical provision [31–33].

Specifically, in PLWHA, peer support interventions have shown to address internalized stigma [34–38] adequately, reduce depressive symptomatology [34] and stress [35–39], enhance the quality of life and wellness [40], and improve treatment adherence [41–48].

Peer support interventions are recommended in official health guidelines such as the Center for Disease and Prevention [49] and the British HIV Association [50].

However, these services are rarely offered in HIV clinics due to existing structural barriers, such as a lack of mental health services and difficulties in accessing services [51, 52]. Against this scenario, internet-based interventions have proliferated. These are easy to access by many people due to resource-saving and flexible delivery [53]. Additionally, internet-based interventions offer anonymity, are easily accessible, and are also scalable [51, 54–56]. Therefore, they have been suggested as an alternative to overcome the barriers mentioned above [57, 58].

Recent reviews of internet-based interventions have significantly impacted outcomes, including adherence, viral load, mental health, and social support for PLWHA [59, 60].

However, none of the reviews has focused on peer support interventions delivered virtually.

We conducted a scoping review to map the existing literature on psychosocial interventions for PLWHA based on peer support and delivered through the internet. We chose the scoping review methodology developed by Peters and colleagues [61] since it allows comprehensive identification of the types and nature of psychosocial interventions for PLWHA, based on peer support and delivered through the internet described in the published literature [61]. Specifically, this review aims to answer the following questions: 1) What internet-based peer support interventions are available for PLWHA? What are their main characteristics? 2) How do the available interventions integrate peer support?

To our knowledge, this is the first systematic effort to provide such an overview.

## Methods

### Protocol and registration

We conducted a systematic scoping review of the peer-reviewed academic literature following the Joanna Briggs Institute (JBI) methodological guidance for scoping review [62] and the Preferred Reporting Items for Systematic Reviews and Meta-Analyses (PRISMA) extension for scoping reviews [63] Our pre-registered protocol containing the detailed methods is available at Open Science Framework (http://osf.io/r729p) (S1 Appendix).

### Eligibility criteria

We included studies about psychosocial interventions designed explicitly for PLWHA and AIDS, based on peer support and delivered through technological devices and/or the internet. For this scoping review, any comparator was relevant for inclusion, and studies without a comparator were also assessed for eligibility. All available publications were eligible for inclusion (e.g., articles -any design, excluding systematic and scoping reviews-articles in conference proceedings, websites, chapters in textbooks). This scoping review had no limitations regarding the time of publication and duration of the intervention and no language restriction. S1 and S2 Appendices show the eligibility criteria and the search strategy.

## Information sources

A comprehensive literature research of electronic bibliographic databases was conducted in PUBMED, Web of Science (WOS), and CINAHL Complete (through EBSCO). This selection was made according to our institutional availability/accessibility; for this reason, some databases were excluded (e.g., EMBASE, Cochrane). All databases and sources of information were consulted on March 10, 2022. The reference lists of 13 relevant reviews on the topic were screened [19, 25, 26, 64–73].

## Search

We developed the search strategy using the PRESS (Peer Review of Electronic Search Strategies) checklist [74], which was adapted to three databases. This step was conducted by the investigators (S.C.C. & A.G.C.) without the collaboration of a librarian due to institutional limitations. The words used were related to telemedicine or internet-delivered interventions (i.e., ehealth, digital health, mobile health), HIV or AIDS, and peer support or support group (for the complete search string, see S3 Appendix) were searched in the articles' title. No other limitation was applied to the search.

## Sources of evidence selection

The database and manual searches were exported into Microsoft Excel [75]. Duplicate papers were removed. Two reviewers (S.C.C. & A.G.C.) independently screened each article for inclusion by title, excluding articles that failed the eligibility criteria. Then, the same two reviewers independently screened the article by abstract using a Google form questionnaire containing details to inform decision-making about inclusion/exclusion. Disagreements between reviewers were resolved through an iterative consensus process involving multiple rounds of deliberative discussion.

## Data charting process

The authors developed a Google form questionnaire with detailed instructions (S.C.C. & A.G.C.) and were approved by the research team to achieve the charting process. This form was guided by the objectives of the present review, being the items related to articles' characterization and their conceptualization of peer support. To ensure internal consistency, some articles were codified in duplicate by two authors (S.C.C. & A.G.C.) and the rest was done independently by the same researchers.

## Data items

First, articles' data were sought regarding study characterization: 1) year of publication; 2) location of the study; 3) study design/article type; 4) population; 5) name of the intervention; and 6) type of technology used. Then, the articles were revised to identify their conceptualization of the peer support component of the intervention (i.e., peer support application).

## Synthesis of results

Data was summarized in a narrative account following the guidelines for scoping reviews [76].

## Results

### Selection of sources of evidence

The initial search yielded 517 articles, and 15 more were found by manual search from reviews' citations. After the removal of duplicate titles, 416 articles were left. Then, two authors (S.C.C. & A.G.C.) screened titles and abstracts, and 28 articles were included in the review and went through the codification process (Fig 1).

### Characteristics of sources of evidence

As shown in Table 1, of the total of included articles, 13 were published during 2017–2019 [77–89], eight during 2020–2022 [90–97], four during the 2014–2016 period [98–101], two were published during 2008–2010 [102, 103], and one during 2011–2013 [104].

The majority of articled revised were studies conducted in the United States (12) [77, 80–82, 84, 88, 90, 94, 95, 99, 100, 104], three were from Kenya [86, 92, 102] and three from South Africa [85, 93, 101]. Locations like China, the United Kingdom and Zambia had two studies included in this review [79, 87, 91, 96, 98, 103] and, from the total of articles included, only one article was from Malaysia, Nigeria, Tanzania and Uganda [78, 83, 89, 97].

Regarding the study type, eight of the included articles corresponded to pilot studies [77, 79, 86, 88, 92, 101, 102, 104], while seven articles explicitly indicate a clinical trial type of design [83, 85, 86, 90, 95, 96, 98], as well as five protocols [80–82, 93, 94] and five qualitative studies [87, 91, 99, 100, 103]. Only two Randomized Controlled Trials [78, 97], and one cohort study [89].

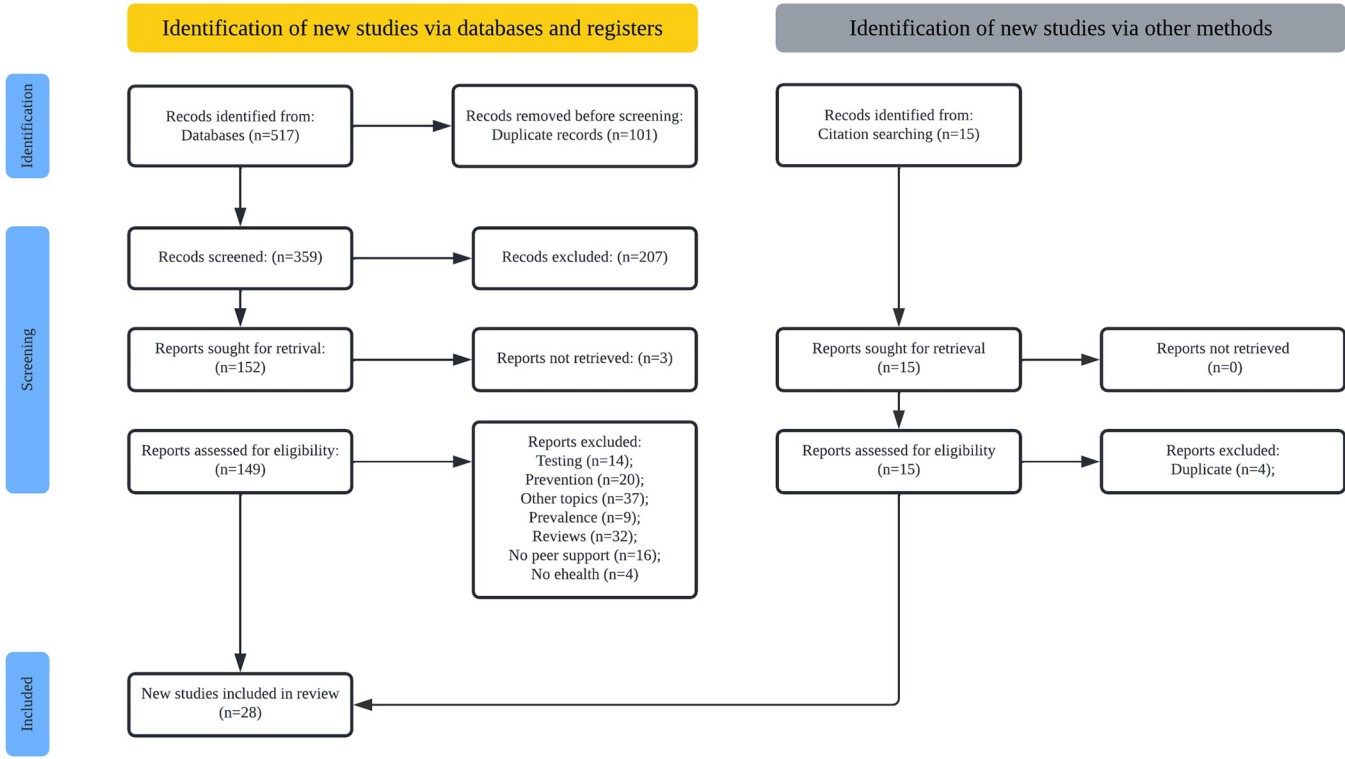

**Fig 1. PRISMA flowchart.**

**Table 1. Characterization of included articles.**

| Articles' Characteristics | |
| --- | --- |
| **Year of Publication** | **n (%)** |
| 2008–2010 | 2 (7.14) |
| 2011–2013 | 1 (3.57) |
| 2014–2016 | 4 (14.28) |
| 2017–2019 | 13 (46.42) |
| 2020–2022 | 8 (28.57) |
| **Location** | |
| China | 2 (7.14) |
| Kenya | 3 (10.71) |
| Malaysia | 1 (3.57) |
| Nigeria | 1 (3.57) |
| South Africa | 3 (10.71) |
| Tanzania | 1 (3.57) |
| Uganda | 1 (3.57) |
| UK | 2 (7.14) |
| USA | 12 (42.85) |
| Zambia | 2 (7.14) |
| **Type of Study** | |
| Pilot[a] | 8 (28.57) |
| Other Clinical Trials[b] | 7 (25) |
| Protocol[c] | 5 (17.85) |
| Qualitative | 5 (17.85) |
| Randomized Clinical Trial | 2 (7.14) |
| Randomized Clinical Trial | 1 (3.57) |

[a] Pilot,feasibility and acceptability trials.

[b] All types of clinical trial designs (e.g. pre-post, with no control group).

[c] Protocols for Randomized Controlled Trials and other designs.

## Synthesis of results

**Peer support interventions.**   Table 2 summarizes the total of interventions included and reviewed in this work. Only 20 of the total of 28 mentioned a specific name for the intervention.

**Interventions' main characteristics.**   The total of interventions included were codified according to their characteristics such as target population, eHealth type and the objective of each intervention. Table 3 summarizes the information of these categories.

*Target population*. All the interventions were exclusively conducted for PLWHA. Of the total of interventions reviewed, 10 of them were orientated to an adult population (18 years or older) [77, 78, 88, 94, 95, 98, 99, 102–104], nine were tailored for children, adolescents, and young adults [79–81, 84–86, 90, 97, 101], three interventions were exclusively designed for adolescents [83, 93, 96], and only one was made exclusive for young adults [100]. Finally, four interventions were orientated to other populations (e.g., mothers, female sex workers, men who have sex with other men [MSM]) [82, 87, 89, 91].

*Type of digital health*. The interventions used a myriad of digital tools to be delivered. The use of social networking platforms such as Facebook and WhatsApp was one of the most frequent strategies (n = 7) of the reviewed studies [79, 83, 85, 92, 93, 100, 101], as well as the use

**Table 2. Interventions' name.**

| Interventions' Name | |
|---|---|
| **Reference** | |
| *Winstead-Derlega et al., 2012* | Positive Project |
| *Broaddus et al., 2015* | My YAP Family |
| *Henwood et al., 2016* | Khaya HIV Positive |
| *Flickinger et al., 2017* | Positive Links |
| *Westergaard et al., 2017* | mPeer2Peer |
| *Dulli et al., 2018* | SMART Connections |
| *Horvath et al., 2018* | Thrive With Me |
| *Hacking et al., 2019* | The Virtual Mentors Program |
| *Horvath et al., 2019* | YouTHrive |
| *Ivanova et al., 2019* | ELIMIKA |
| *Knudson et al., 2019* | China MP3 (Multi-component HIV Intervention Packages for Chinese MSM) |
| *Navarra et al., 2019* | ACCESS (Adherence Connection for Counseling, Education, and Support) |
| *Tun et al., 2019* | CBHTC+ (Intervention within Sauti project) |
| *Hay et al., 2020* | 4MNetwork |
| *MacCarthy et al., 2020* | SITA (SMS as an Incentive To Adhere) |
| *Ochoa et al., 2021* | LINX App / LINX App Plus |
| *Simpson et al., 2021* | Insaka |
| *Steinbock et al., 2022* | End+dDisparities ECHO Collaborative |
| *Stockman et al., 2021* | LinkPositively |
| *Zanoni et al., 2022* | InTSHA (Interactive Transition Support for Adolescents Living With HIV using Social Media) |
| *Mo & Coulson, 2008* | Unnamed |
| *Wools-Kaloustian et al., 2009* | |
| *Mi et al., 2015* | |
| *Abdulrahman et al., 2017* | |
| *Senn et al., 2017* | |
| *Rotheram et al., 2019* | |
| *St Clair-Sullivan et la., 2019* | |
| *Chory et al., 2022* | |

of websites (n = 7) [80–82, 86, 98, 102, 103]. The development of a smartphone App was also present in 4 interventions [77, 88, 96, 99], along with the use of SMS and phone calls to establish communication between peers [78, 81, 85, 93]. Only three studies developed a web-based platform [84, 94, 95], two used SMS communication exclusively [87, 97], and two interventions were delivered via videoconference [90, 98]. Only one intervention showed the participants´ videos made by peers [104].

*Interventions' goals.* The most common objectives were treatment adherence (n = 14) [78, 80, 82–86, 88, 89, 92, 95, 99, 102, 104], and social support (n = 7) [77, 83, 91, 94, 96, 101, 103]. Five of the interventions were dedicated to retention in care [87, 88, 93, 95, 101], and four to viral load suppression [78, 80, 90, 95]. Two interventions were oriented to stigma reduction [92, 104] and the other two to increase HIV knowledge [83, 98]. Finally, the aims of well-being [100], mental health [92], and legal support [94] were included only once each.

**Peer support implementation.** The role of peer support was incorporated differently in the revised interventions. Some interventions combined more than one strategy to implement

**Table 3. Main characteristics of the included interventions.**

| Interventions' Main Characteristics | | | |
|---|---|---|---|
| Reference | Target Population | Digital Health Tool | Interventions' Objective |
| *Winstead-Derlega et al., 2012* | Rural adults (18 or older) | iPod preprogrammed with peer health videos | Improve treatment adherence and reduce the perception of stigma |
| *Broaddus et al., 2015* | Young adults (16–25 years) | Private Facebook group | Improve patient well-being |
| *Henwood et al., 2016* | Adolescents and young adults (12–25 years) | Chat-room through MXit social networking platform | Retain youth throughout the continuum of care and provide ongoing social support within a peer learning environment |
| *Flickinger et al., 2017* | Adults (18 or older), attending a university clinic | Smartphone App | Improve treatment adherence |
| *Westergaard et al., 2017* | Adults (18 or older), history of substance abuse | Smartphone App | Support HIV treatment for patients who had been marginally engaged in care |
| *Dulli et al., 2018* | Adolescents (15–19 years), on ART treatment | Private Facebook group | Improve HIV knowledge, social support, and treatment adherence |
| *Horvath et al., 2018* | Men (MSM[d]), suboptimal adherence to treatment | Website and SMS | Assess the impact of the intervention on the target population |
| *Hacking et al., 2019* | Adolescents and young adults (12–25 years), newly diagnosed HIV positive, not in treatment | Smartphone communication (SMS, phone call or WhatsApp) | Improve treatment adherence by referring patients to an adherence club |
| *Horvath et al., 2019* | Adolescents and young adults (15–24 años) | Website | Enhance treatment adherence and improve other outcomes (e.g. decreased viral load) |
| *Ivanova et al., 2019* | Adolescents and young adultos (15–24 years), all level of treatment | Website | Improve treatment adherence |
| *Knudson et al., 2019* | Men (MSM) newly diagnosed HIV positive | SMS | Facilitate engagement in care and initiation of antiretroviral therapy |
| *Navarra et al., 2019* | Adolescents and young adults (16–29 years), belonging to ethnic minority (African Americans and Hispanics/Latinos) | Mobile platform | Improve treatment adherence |
| *Tun et al., 2019* | Female sex workers (FSW) | WhatsApp | Improve treatment adherence |
| *Hay et al., 2020* | Mothers | WhatsApp | Improve social support (informational, emotional, and practical support) |
| *MacCarthy et al., 2020* | Adolescents and young adults (15–24 years), taking ART | SMS | Improve treatment adherence |
| *Ochoa et al., 2021* | Adults (18 or older), male Black or African American, belonging to a sexual minority | Web based mobile App | Provide social and legal resources and peer support |
| *Simpson et al., 2021* | Adolescent pregnant women (28–34 weeks of pregnancy) | Smartphone (message platform) | Assess the feasibility and acceptability of this mobile phone-based support group intervention |
| *Steinbock et al., 2022* | Adolescents and young adults (13–24 years), men (MSM) with men of color, Black/African American and Latina women, and transgender people | Videoconferences | Improve rates of viral suppression |
| *Stockman et al., 2021* | Adults (18 or older) Woman with African American, Black, or of African descent and experience of interpersonal violence | Web based App | Improve retention in care, treatment adherence, and viral suppression |
| *Zanoni et al., 2022* | Adolescents (15–19 years), with perinatally acquired HIV | Smartphone (websites, phone call, WhatsApp) | Evaluate the retention in care during the transition from pediatric to adult care |
| *Mo & Coulson, 2008* | Adults | Website | Improve social support |
| *Wools-Kaloustian et al., 2009* | Adults, stable in cART treatment | Website | Improve treatment adherence |
| *Mi et al., 2015* | Adults (18 or older), men (MSM) | Website, online sessions (discussion and counseling) | Promote safe sex behaviors and access to HIV services |
| *Abdulrahman et al., 2017* | Adults | SMS, phone call | Enhance treatment adherence and improve other outcomes (e.g. decreased viral load) |
| *Senn et al., 2017* | Adults (18 or older), black men (MSM) | Smartphone App | Improve retention in care and treatment adherence |

*(Continued)*

**Table 3.** (Continued)

| | Interventions' Main Characteristics | | |
| --- | --- | --- | --- |
| Reference | Target Population | Digital Health Tool | Interventions' Objective |
| *Rotheram et al., 2019* | Adolescents and young adults, all level of treatment | Website, SMS, and phone call | Promote retention in care during treatment continuum in youth |
| *St Clair-Sullivan et la., 2019* | Adolescents and young adults (16–24 years), currently receiving HIV care | Smartphone communication (WhatsApp and Facebook) | Identify barriers to HIV care and the acceptability and of mHealth to improve treatment adherence |
| *Chory et al., 2022* | Children and adolescents (10–19 years), on ART treatment | WhatsApp | Enhance treatment adherence, reduce stigma and improve mental health |

[d]Men who have Sex with other Men.

peer support. The communication via posts in a group board or online forums was one of the most common interventions (n = 7) [79–82, 94, 99, 103], followed by the use of peer counselors (n = 5) [78, 81, 84, 87, 98] and the use of SMS or WhatsApp to establish contact between peers (n = 5) [81, 88, 91–93]. Implementing trained peers to provide psychosocial and logistical support was also a strategy for four of the revised interventions [77, 85, 86, 102], and the use of online support groups was also frequently presented in the interventions [83, 90, 100, 101]. Two of the reviewed studies used online focus groups [96, 97], one implemented peer education [89], and only one used videos made by peers [104]. At last, one intervention generated a strategy of matched peers who had similar trauma experiences [95]. As an important component, three of the reviewed interventions incorporated peer support anonymously [79, 96, 99]. Table 4 summarizes the type of peer support implemented by each intervention.

## Discussion

### Summary

This review aimed to systematically scope the empirical literature on peer-support psychosocial interventions for PLWHA. More specifically, we aimed to 1) identify the existent digital peer support interventions currently available for PLWHA; 2) summarize the main characteristics of the available interventions 3) examine how the interventions implemented peer support in a virtual environment.

Twenty-eight studies were identified in a systematic search across peer-reviewed journals.

Papers were primarily pilot studies and protocols published in North America or Africa within the last 5 years. This recent increase in papers reflects the growing interest in developing peer-support eHealth interventions for PLWHA. Even though only three studies were RCT, the relatively large number of RCT protocols suggests that this field will continue growing in the coming years. Participants were mainly HIV+ adults predominantly from minority ethnic, racial and/or sexual backgrounds. None of the studies was conducted in Latin America, which is problematic considering the high prevalence of HIV (approximately 1.8 million people in 2017) [105, 106], the difficulties presented in achieving the 90-90-90 targets designated by UNAIDS [107], and the tendency for late treatment initiation [105].

Social networks and messaging apps (such as Facebook or WhatsApp) were the most frequently used digital health tools, which is consistent with research suggesting the increasing validity of psychosocial interventions using social networks for different populations [108–110]. Considering the ongoing massification of both smartphones [111–113] and access to the internet worldwide [114, 115], this is a positive finding and suggests that there are indeed eHealth interventions that could be more widely accessed.

**Table 4. Description of how the peer support was implemented.**

| Peer Support Implementation | |
|---|---|
| **Reference** | **Peer Support Type** |
| *Winstead-Derlega et al., 2012* | Peer messages delivered through videos |
| *Broaddus et al., 2015* | Online support groups |
| *Henwood et al., 2016* | Online and face to face support groups |
| *Flickinger et al., 2017* | Interaction through a community message board (CMB) with anonymous usernames |
| *Westergaard et al., 2017* | Peer trained to deliver intensive psychosocial and logistical support |
| *Dulli et al., 2018* | Support groups moderated by trained peers |
| *Horvath et al., 2018* | Online forum, social network posts |
| *Hacking et al., 2019* | Peer as trained mentees that contact recently diagnosed people to attend an adherence club |
| *Horvath et al., 2019* | Online forum, messages and social network posts |
| *Ivanova et al., 2019* | Peer as trained mentees that contact diagnosed people to participated in an adherence intervention |
| *Knudson et al., 2019* | Face to face counseling and contact with via SMS |
| *Navarra et al., 2019* | Peers trained as coaches |
| *Tun et al., 2019* | Peer education |
| *Hay et al., 2020* | WhatsApp groups |
| *MacCarthy et al., 2020* | Focus group |
| *Ochoa et al., 2021* | Online forum |
| *Simpson et al., 2021* | Focus group, first interviews and SMS communication |
| *Steinbock et al., 2022* | Online support group |
| *Stockman et al., 2021* | Match with a trained and trauma-informed virtual peer, communication via smartphone |
| *Zanoni et al., 2022* | WhatsApp groups |
| *Mo & Coulson, 2008* | Messages posted at an online board |
| *Wools-Kaloustian et al., 2009* | Instructors that mediates between medical attention and patients giving advices |
| *Mi et al., 2015* | Online peer counseling and giving information via website |
| *Abdulrahman et al., 2017* | Online peer counseling |
| *Senn et al., 2017* | SMS texting with participants |
| *Rotheram et al., 2019* | Social media forums and coaching via SMS, phone, or in-person |
| *St Clair-Sullivan et la., 2019* | Online support forum |
| *Chory et al., 2022* | WhatsApp groups |

The most common peer activity was the participation in social networking posting, peer counseling, and peer discussions and conversations through WhatsApp or other social messaging services, which are considered to be an asynchronous form of technology [116].

Interestingly, very few interventions [78, 96–98] incorporating face-to-face synchronic interaction were identified. Even though numerous studies have shown that synchronous technologies (such as real-time video conferencing) are a valid method to deliver group psychosocial interventions [116, 117], real-time activities present constraints (i.e., scheduling) that can be overcome with asynchronous technologies[116].

Also, digital support emerges as a promising approach to complement healthcare [118, 119]. For instance, through digital peer support, patients may have more efficient access to both health care services and HIV-related information (e.g., whether and how often the person should seek medical assistance based on symptoms).

It is worth noting that although internet-based interventions may help ease difficulties in access for some PLWHA—access to these interventions may be limited for some populations and marginalized groups (e.,g., older people, people with severe mental health conditions, people with specific disabilities,) [120–122]. Likewise, the risk of digital exclusion may make a strong point for face to face services.

## Limitations

Our scoping review has two main limitations. Firstly, it was conducted only in 3 databases (PUBMED, Web of Science, and CINAHL Complete). The selection of these databases was due to a limited institutional budget; for this reason, some databases were excluded (e.g., EMBASE, Cochrane).

Secondly, and also due to institutional limitations, we did not count with the collaboration of a librarian, which may have had an impact on the expertise in designing and refining the main gsearch of our paper.

## Conclusion

In this review we have summarized the digital peer support interventions currently available for PLWHA, their main characteristics, and the way in which they implemented peer support in a virtual environment.

Overall the reported outcomes appeared promising, especially regarding potential improvements in treatment adherence and enhanced perceived social support. Future research should focus on continuing collecting data through RCTs studies in diverse social contexts. Having robust diverse evidence of the effectiveness of this type of interventions may help expand the scope and the impact of different treatments.

## Supporting information

**S1 Appendix. Pre-registration protocol at open science framework.** Protocol developed by the researchers following the Open Science Framework guidelines.
(DOCX)

**S2 Appendix. Eligibility criteria.** List of the eligibility criteria used to assess the articles for inclusion.
(DOCX)

**S3 Appendix. String of search.** Full string of search implemented in PUBMED. The string of search was adapted to each database.
(DOCX)

**S4 Appendix. Preferred Reporting Items for Systematic reviews and Meta-Analyses extension for Scoping Reviews (PRISMA-ScR) checklist.** Checklist completed by the researchers following PRISMA guidelines.
(DOCX)

## Author Contributions

**Conceptualization:** Stefanella Costa-Cordella, Aitana Grasso-Cladera, Javiera Duarte, Claudia P. Cortes.

**Data curation:** Stefanella Costa-Cordella.

**Formal analysis:** Stefanella Costa-Cordella, Aitana Grasso-Cladera.

**Funding acquisition:** Alejandra Rossi, Claudia P. Cortes.

**Investigation:** Stefanella Costa-Cordella, Aitana Grasso-Cladera.

**Methodology:** Stefanella Costa-Cordella, Aitana Grasso-Cladera, Javiera Duarte.

**Project administration:** Stefanella Costa-Cordella.

**Resources:** Stefanella Costa-Cordella.

**Supervision:** Stefanella Costa-Cordella, Alejandra Rossi, Claudia P. Cortes.

**Validation:** Stefanella Costa-Cordella.

**Visualization:** Stefanella Costa-Cordella, Aitana Grasso-Cladera.

**Writing – original draft:** Stefanella Costa-Cordella, Aitana Grasso-Cladera.

**Writing – review & editing:** Stefanella Costa-Cordella, Aitana Grasso-Cladera, Alejandra Rossi, Javiera Duarte, Flavia Guiñazu, Claudia P. Cortes.

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
