## [Decision Letter · Decision Letter 0]

15 Feb 2022

PONE-D-21-30570

Internet-based peer support interventions for people living with HIV: A Scoping Review

PLOS ONE

Dear Dr. Costa-Cordella,

Thank you for submitting your manuscript to PLOS ONE. After careful consideration, we feel that it has merit but does not fully meet PLOS ONE’s publication criteria as it currently stands. Therefore, we invite you to submit a revised version of the manuscript that addresses the points raised during the review process.

We look forward to receiving your revised manuscript.

Kind regards,

Bronwyn Myers

Academic Editor

PLOS ONE

https://journals.plos.org/plosone/s/fileid=ba62/PLOSOne_formatting_sample_title_authors_affiliations.pdf".

“This work was funded by FONDEF ID20I10174, Agencia Nacional de Investigación y Desarrollo (Chile) and the Agencia Nacional de Investigación y Desarrollo (ANID, Chile) through FONDECYT regular N° 1190610.”

“This study was funded by the Chilean National Agency of Research and Development (Agencia Nacional de Investigación y Desarrollo de Chile) through FONDEF to CC (ID20I10174),  and the Chilean National Agency of Research and Development (Agencia Nacional de Investigación y Desarrollo de Chile)  through FONDECYT to AR (N° 1190610). The funders had no role in study design, data collection and analysis, decision to publish, or preparation of the manuscript.”

Additional Editor Comments:

Thank you for submitting this manuscript for review. We now have two reviewer's reports- both of which recommend significant modifications to the methods- this will require a new search strategy and rerunning your searches. It will also require you to follow reporting guidelines and frameworks for scoping reviews. Both reviewers provide extensive comments that I hope you will find helpful as you revise the manuscript.

Reviewers' comments:

Reviewer's Responses to Questions

**Comments to the Author**

1. Is the manuscript technically sound, and do the data support the conclusions?

Reviewer #1: Partly

Reviewer #2: No

2. Has the statistical analysis been performed appropriately and rigorously? 

Reviewer #1: N/A

Reviewer #2: N/A

3. Have the authors made all data underlying the findings in their manuscript fully available?

Reviewer #1: No

Reviewer #2: Yes

4. Is the manuscript presented in an intelligible fashion and written in standard English?

Reviewer #1: Yes

Reviewer #2: No

5. Review Comments to the Author

Reviewer #1: Thank you for the opportunity to review this manuscript. It addresses an important context. I provide recommendations for improvement that could strengthen its relevance and impact

Introduction:

Paragraph 2. The authors fail to consider the impact of violence and even more importantly alcohol and other drug use on HIV adherence. Substance use stigma also intersects with HIV stigma to worsen outcomes A variety of interventions have been developed to address these issues to improve likelihood of adherence. The literature review should be expanded to reflect this. Some articles of relevance include:

Regenauer, K.S., Myers, B., Batchelder, A.W., Magidson, J.F. (2020). “That person stopped being human”: Intersecting HIV and substance use stigma among patients and providers in South Africa. Drug and Alcohol Dependence, 216, 108322. https://doi.org/10.1016/j.drugalcdep.2020.108322.

Myers, B., Lombard, C., Joska, J.A. Abdullah, F., Naledi, T., Lund, C., Petersen Williams, P., Stein, D.J., Sorsdahl, K.R. Associations Between Patterns of Alcohol Use and Viral Load Suppression Amongst Women Living with HIV in South Africa. AIDS Behav (2021). https://doi.org/10.1007/s10461-021-03263-3

Abrahams N, Mhlongo S, Dunkle K, Chirwa E, Lombard C, Seedat S, Kengne AP, Myers B, Peer N, Garcia-Moreno C, Jewkes R. Increase in HIV incidence in women exposed to rape. AIDS. 2021 Mar 15;35(4):633-642. doi: 10.1097/QAD.0000000000002779

Relatedly, Some additional interventions to address psychosocial risks to adherence (specifically related to alcohol and other drugs) should be mentioned:

Zule, W., Myers, B., Carney, T., Novak, S., McCormick, K., & Wechsberg, W.M. (2014). Alcohol and drug use outcomes among vulnerable women living with HIV: results from the Western Cape Women’s Health CoOp. AIDS Care 26: 1494-9.

Belus, J., Rose, A.I., Anderson, L., Ciya, N., Joska, J., Myers, B., Safren, S.A., Magidson, J. (2020). Adapting a behavioral intervention for alcohol use and HIV medication adherence for lay counselor delivery in Cape Town, South Africa. A case series. Cognitive Behavioral Practice. https://doi.org/10.1016/j.cbpra.2020.10.003.

Magidson, J. F., Joska, J. A., Belus, J. M., Andersen, L. S., Regenauer, K. S., Rose, A. L., Myers, B., Majokweni, S., O’Cleirigh, C. and Safren, S. A. Project Khanya: Results from a pilot randomized type 1 hybrid effectiveness-implementation trial of a peer-delivered behavioural intervention for ART adherence and substance use in HIV care in South Africa. J Int AIDS Soc. 2021; 24(S2):e25720.

Paragraph 3. There has been some other work around peer support interventions for HIV, including a recent systematic review that is worth acknowledging. These do not offer internet/digital interventions so worth distinguishing this previous review from the current one. Please see below

Satinsky, E.N., Kleinman, M.B., Tralka, H.R., Jack, H.E., Myers, B., Magidson, J.F. (2021). Peer-delivered services for substance use in low- and middle-income countries: A systematic review. International Journal of Drug Policy, 95, 103252. https://doi.org/10.1016/j.drugpo.2021.103252

Magidson, J. F., Joska, J. A., Belus, J. M., Andersen, L. S., Regenauer, K. S., Rose, A. L., Myers, B., Majokweni, S., O’Cleirigh, C. and Safren, S. A. Project Khanya: Results from a pilot randomized type 1 hybrid effectiveness-implementation trial of a peer-delivered behavioural intervention for ART adherence and substance use in HIV care in South Africa. J Int AIDS Soc. 2021; 24(S2):e25720.

Magidson, J.F., Joska, J., Regenauer, K.S., Satinsky, E., Andersen, L., Seitz-Brown, C.J., Borba, C.P.C., Safren, S.A., Myers, B. (2019). “Someone who is in this thing that I am suffering from”: The role of peers and other facilitators for task sharing substance use treatment in South African HIV care." International Journal of Drug Policy 70: 61-69.

Paragraph 5. It is worth noting that although internet-based interventions may help ease difficulties in access for some PLWH- access to these interventions may be limited for economically deprived populations and marginalised groups. – ie there is a risk of digital exclusion- so these could be an adjunct for face to face services. There is a large literature now from COVID-19 mental health services that highlights the risk of digital exclusion in both high and lower income settings. This is especially important given that poorer countries still have a higher prevalence of HIV. I recommend expanding on this caveat in the discussion section too.

Final paragraph of the introduction: I think you mean to write the “no specificity or broadness of the research question rather than specificity- please revise accordingly.

Methods

1. Did you follow a scoping review framework- eg Joanna Briggs?

2. I am not sure what is meant by “lack of extension in the area of interest- please clarify (search strategy)

Results

1. Intervention goals- it is worth noting that none of these interventions addressed underlying psychosocial factors known to adversely impact adherence and outcomes (e.g mental health, AOD use, violence)

2. Study findings- how was treatment adherence measured? Confidence in taking medication is a weak proxy for adherence? Any impact on viral load? My takeaway would be the need for more robust studies with more objective measures of adherence and HIV outcomes

Discussion: I think you can make a stronger statement about the lack of robust designs, the heterogeneity of the approaches and the need for RCTs that provide evidence (objective) of the impact of these kinds of interventions on HIV treatment engagement, ART use and viral load. Also need to demonstrate that these approaches offer add on value beyond face to face interventions.

I think you need to soften your statements about the findings being in line with what is known about the effectiveness of face to face peer interventions. At best the studies included in your review suggest these are feasible and acceptable interventions and show some promise but the field is nascent and more evidence is needed.. The discussion would benefit from a review of the quality of the included studies so as to identify next steps for research

Reviewer #2: I appreciate the opportunity to review this manuscript and read it with great interest. The authors are to be commended for having undertaken this work. Regrettably, the methods have several weaknesses, which raise considerable doubts about the internal validity of the results of the review. However, if the methods of the review are improved such that the results have internal validity, the results could be trusted and would be of interest to readers.

I find that this is not a well conducted scoping review. Methodological guidelines for conducting scoping reviews exist, and to earn the distinction ‘scoping review’, these guidelines should be followed. The authors state that they used PRISMA for scoping reviews, but this is a reporting tool, not a methods guide. Still, the PRISMA items help show what is missing to make this a review with results that readers can trust. For example, the authors fail to: Specify characteristics of the sources of evidence used as eligibility criteria and provide a rationale; Present the full electronic search strategy for at least 1 database, including any limits used, such that it could be repeated. The authors refer to a protocol, but the ‘protocol’ only includes a table (the same as table 2 in the manuscript) and no description of pre-specified review methods. Table 2 in the manuscript describes population, exposure, comparison, outcomes, study design, which is what one would expect to find in a review about effect. This is not a review about effect, thus this demonstrates the researchers’ unfamiliarity with the methods of scoping reviews. Additionally, they excluded family publications derived from the same study, and included only “the study with the most robust evidence, i.e. RCT” (page 11), which gives a distorted (incomplete) presentation of evidence on internet-based peer support interventions. RCT is a robust design for investigating questions about effect, but, again, the researchers did not conduct a review about effect. It is strange that the researchers decided to use the Popay et al (2006) narrative synthesis method, and not the methods described in guidance publications on scoping reviews, e.g. Arksey and O’Malley 2005; Levac et al 2010; Armstrong et al 2011; Peters et al 2015.

The literature search appears to be neither exhaustive nor systematic (see e.g. PRESS checklist). It is strange that EMBASE was not searched. Table 1, which shows search terms, reveals that the search strategy was poor and not systematic and thus cannot be viewed as ensuring all eligible studies were identified. This is demonstrated by the fact that only 234 records were identifies in the database searches. This is several thousand fewer identified records than similar systematic reviews on the same topic. The eligibility criteria are inadequate, missing specification of e.g. study participants and definition/operationalization of the exposure. How was digital health understood? How was peer support understood? Given the central exposure in the review is peer support intervention, and the authors repeatedly refer to “peer support components”, it is a weakness that they do not specify peer support. The Peers for Progress program identify four key functions, or components, for peer-support: assistance in daily management, social and emotional support, linkage to clinical care and community resources, and ongoing support related to chronic disease. It is questionable whether the nine included interventions in fact are peer support interventions. Another underlying premise of peer-support is personalized interaction, which is difficult to achieve with digital communication and therefore needs to be addressed here.

Additionally, I find that the researchers’ rational for a review on internet-based peer support is weak (they write for example that there is a “lack of extension of the area of interest” page 10), their categories in Table 3 are not consistent (e.g. they specify ‘pilot’ as a study design, ‘study location’ is a mix of countries and cities). When they report on ‘study findings’ they mix study designs such as RCTs and qualitative studies when reporting on effect, when, obviously, qualitative studies do not assess effects. In the discussion section, they make claims about mechanisms that support the interventions, when there is no information about mechanisms in the results section. Similarly, in the discussion section, they discuss anonymity as an advantage of internet-based peer support interventions, when there is no information about anonymity in the results section. In the limitations, they mention exclusions that are not mentioned in the methods section and they state that their inability to conduct meta-analyses is a limitation, which is completely irrelevant because they (state they) conducted a scoping review. Lastly, I find that the review is poorly written, marred with language errors, and the unclear language impedes understanding.

6. PLOS authors have the option to publish the peer review history of their article (what does this mean?). If published, this will include your full peer review and any attached files.

Reviewer #1: No

Reviewer #2: No

---

## [Author Response · Author response to Decision Letter 0]

12 Apr 2022

Response to Reviewer 1:

Thank you for your review of our paper. We have answered each of your points below. (See cover letter-response to reviewer).

Response to Reviewer 2:

Dear reviewer 2,

We would like to thank you for your detailed feedback. We have significantly improved our knowledge

regarding the development of scoping reviews and we have re-conducted every step of the process

again.

Please find the details of the changes below.(See cover letter-response to reviewer).

---

## [Decision Letter · Decision Letter 1]

19 May 2022

Internet-based peer support interventions for people living with HIV: A Scoping Review

PONE-D-21-30570R1

Dear Dr. Costa-Cordella,

We’re pleased to inform you that your manuscript has been judged scientifically suitable for publication and will be formally accepted for publication once it meets all outstanding technical requirements.

Kind regards,

Bronwyn Myers

Academic Editor

PLOS ONE

Additional Editor Comments (optional):

Reviewers' comments:

Reviewer's Responses to Questions

**Comments to the Author**

1. If the authors have adequately addressed your comments raised in a previous round of review and you feel that this manuscript is now acceptable for publication, you may indicate that here to bypass the “Comments to the Author” section, enter your conflict of interest statement in the “Confidential to Editor” section, and submit your "Accept" recommendation.

Reviewer #1: All comments have been addressed

2. Is the manuscript technically sound, and do the data support the conclusions?

Reviewer #1: (No Response)

3. Has the statistical analysis been performed appropriately and rigorously? 

Reviewer #1: (No Response)

4. Have the authors made all data underlying the findings in their manuscript fully available?

Reviewer #1: (No Response)

5. Is the manuscript presented in an intelligible fashion and written in standard English?

Reviewer #1: (No Response)

6. Review Comments to the Author

Reviewer #1: (No Response)

7. PLOS authors have the option to publish the peer review history of their article (what does this mean?). If published, this will include your full peer review and any attached files.

Reviewer #1: No

---

## [Editor Report · Acceptance letter]

19 Aug 2022

PONE-D-21-30570R1 

Internet-based peer support interventions for people living with HIV: A Scoping Review 

Dear Dr. Costa-Cordella:

I'm pleased to inform you that your manuscript has been deemed suitable for publication in PLOS ONE. Congratulations! Your manuscript is now with our production department. 

Kind regards, 

on behalf of

Dr. Bronwyn Myers 

Academic Editor

PLOS ONE